# The spindle assembly checkpoint and the spatial activation of Polo kinase determine the duration of cell division and prevent tumor formation

Emmanuel Gallaud[1], Laurent Richard-Parpaillon[1], Laetitia Bataillé[1], Aude Pascal[1], Mathieu Métivier[1], Vincent Archambault[2], Régis Giet[1]*

**1** Univ Rennes, CNRS, INSERM, IGDR (Institut de Génétique et Développement de Rennes) UMR 6290, ERL U1305, Rennes, France, **2** Institute for Research in Immunology and Cancer, Université de Montréal, Montréal, Quebec, Canada

* regis.giet@univ-rennes1.fr

**Data Availability Statement:** All relevant data are within the manuscript and its Supporting Information files.

## Abstract

The maintenance of a restricted pool of asymmetrically dividing stem cells is essential for tissue homeostasis. This process requires the control of mitotic progression that ensures the accurate chromosome segregation. In addition, this event is coupled to the asymmetric distribution of cell fate determinants in order to prevent stem cell amplification. How this coupling is regulated remains poorly described. Here, using asymmetrically dividing *Drosophila* neural stem cells (NSCs), we show that Polo kinase activity levels determine timely Cyclin B degradation and mitotic progression independent of the spindle assembly checkpoint (SAC). This event is mediated by the direct phosphorylation of Polo kinase by Aurora A at spindle poles and Aurora B kinases at centromeres. Furthermore, we show that Aurora A-dependent activation of Polo is the major event that promotes NSC polarisation and together with the SAC prevents brain tumor growth. Altogether, our results show that an Aurora/Polo kinase module couples NSC mitotic progression and polarisation for tissue homeostasis.

## Author summary

Stem cells divide asymmetrically and exist in limited numbers in tissues to ensure their development and repair. To do so, stem cells are subjected to a rigorous progression of mitotic events to avoid chromosome segregation errors during cell division, and formation of aneuploid daughter cells with impaired proliferative capacities. Moreover, during mitosis, these stem cells are characterized by their polarization along an apical-basal axis. The orientation of the mitotic spindle along this axis allows the asymmetric distribution of cell fate determinants after cell division. This differential segregation is required to contain stem cell amplification and tumor formation. How mitotic progression is coupled to cell polarization at the molecular level during stem cell asymmetric division is therefore a critical question. We show here that the levels of active Polo kinase regulate the duration of cell division. Polo activation is regulated via a direct activating phosphorylation by

**Funding:** Fondation pour la Recherche Médicale (FRM):DEQ20170336742 to RG. Ligue Contre le Cancer to RG. Canadian Institutes of Health Research to VA. The funders had no role in study design, data collection and analysis, decision to publish, or preparation of the manuscript.

**Competing interests:** The authors declare that they have no conflict of interest.

Aurora A and Aurora B kinases. Moreover, activation of Polo kinase by Aurora A is essential to maintain polarization of mitotic stem cells and consequently to prevent tumor formation. Altogether, our results strongly suggest that an Aurora-dependent phosphorylation cascade leading to the activation of Polo kinase regulates the coupling between mitotic progression and tissue homeostasis.

## Introduction

The maintenance of a restricted number of stem cells is essential for tissue homeostasis and repair [1]. This process has been particularly well described during *Drosophila* brain development; loss of NSCs causes microcephaly while their amplification triggers tissue overgrowth [2–6]. NSCs are subjected to asymmetric cell division. This event is characterized by the asymmetric localization of PAR proteins at the apical cortex that triggers cell fate determinants targeting at the basal cortex. After mitotic spindle alignment along the apico-basal axis, NSC asymmetric cell division gives rise to one cell retaining the NSC proliferation fate while the other cell (GMC for Ganglion Mother Cell) inherits cell fate determinants and is subjected to differentiation (reviewed in [7]). Both AurA and Polo are needed for cell polarization and for mitotic spindle alignment [8–11]. As a consequence, in *aurA* and *polo* mutants, the two daughter cells acquire the NSC proliferative fate, resulting in the amplification of the NSC population and tumor formation in neural tissues. Therefore, in the context of central brain development, *aurA* and *polo* genes are both tumor suppressors [2,9,10,12].

Another remarkable feature of *aurA* mutant neural tissue is the absence of obvious chromosome segregations defects despite severe impairment of mitotic spindle assembly [2]. Correct chromosome segregation in many cell types depends on the Spindle Assembly Checkpoint (SAC) that delays mitotic progression and anaphase onset until all kinetochores are properly attached to spindle microtubules (reviewed in [13]). As a consequence, ablation of SAC genes in dividing cancer cells induces premature anaphase onset and chromosome segregation defects, cell death or premature differentiation [2,3,14–17]. *aurA* mutant NSCs show defective mitotic spindle assembly and Cyclin B-Cdk1 levels persist longer than normal, consistent with SAC-dependent mitotic delay [2,18,19]. However, SAC ablation in *aurA* mutant does not restore normal mitotic progression, indicating that AurA regulates mitotic progression independently from the SAC [2]. Altogether, these studies suggest that tissue homeostasis is regulated by AurA kinase that couples mitotic progression to accurate asymmetric cell division of NSCs. The possible AurA targets required for these processes remains elusive. Polo-like kinases 1 were shown, in some cases, to be activated by Aurora kinases *in cellulo* [20–25]. Polo was therefore a potential candidate to function downstream of Aurora A in NSCs. In this study we analyzed asymmetric cell division and mitotic progression in *polo*, *aurA* and *aurB/ial* mutants using gain and loss-of function alleles. In the context of the stem-cell based *Drosophila* central brain development, we show here that asymmetric cell division is coupled to mitotic progression and mainly regulated by the direct activation of Polo kinase by Aurora A and Aurora B *in vivo*.

## Result

### *polo* mutants NSCs exhibit a SAC-independent mitotic delay caused by defective Cyclin B degradation

We first wished to determine if Polo activity promotes mitotic progression in NSCs. To avoid delay caused by possible SAC activation, we examined how time in mitosis was regulated in

different *polo* mutants associated with a loss of the SAC gene Mad2 (*polo,mad2*[P]). Western blotting analyses of brain tissues confirmed the absence of Mad2 in the double mutants and the lower or absent Polo protein levels in weak and strong hypomorphs respectively (Fig 1A). Using live-imaging microscopy, we found that the weak *polo*[1]/*polo*[10] and *polo*[1]/*polo*[9] hypomorphic mutants showed a prolonged mitosis that was not decreased by absence of the SAC (Fig 1B and 1C). The time in mitosis was even more prolonged in strong hypomorphic *polo*[9]/ *polo*[10] mutants but by contrast to weak *polo* hypomorphs, ablation of the SAC decreased the time in mitosis. It indicates a partial SAC contribution to mitotic delay in strong hypomorphic *polo* mutants. This trends were confirmed by analyses of the delay between SAC silencing (as detected by the loss of GFP::Mad2 at kinetochores) and anaphase onset in weak and strong *polo* mutants (Fig 1D and 1E). As a positive control, NSCs were treated by colchicine to activate the SAC. Under these conditions, GFP::Mad2 signal on the kinetochores persisted during all the duration of the experiment, confirming that the SAC was functional in NSCs [26]. The time between SAC silencing and anaphase onset was 0.8 min in controls, 2.1 to 2.4 min in *polo*[1]/*polo*[10] and *polo*[1]/*polo*[9] and 8.6 min in *polo*[9]/*polo*[10] NSCs. This result reveals that weak interference with Polo kinase prolonged mitosis mostly in a SAC-independent manner while mitotic delay is partly SAC-dependent in strong *polo* hypomorphs. Altogether, these results indicate that the SAC cannot account for the prolonged mitosis of weak and strong *polo* hypomorphic mutant NSCs. In a previous study, we showed that a similar SAC-independent mitotic delay was caused by a failure to timely degrade Cyclin B in *aurA* mutants [2]. We therefore analyzed Cyclin B degradation in strong hypomorphic *polo* mutants lacking a functional SAC by adding the *mad2*[P] null mutation [26]. We found that the mitotic delay of *polo* mutants was accompanied by an impaired Cyclin B degradation (S1 Fig). These phenotypes were partly rescued by abrogation of the SAC. Therefore, *aurA* and *polo* mutant NSCs both show a SAC-independent Cyclin B degradation causing a mitotic delay.

## Thresholds of Polo activation control mitotic progression

In human cultured cells and *Xenopus*, the activation of the Polo orthologue Plk1 was shown to be triggered by the phosphorylation of the conserved Thr210 residue and Plk1 activity is sensitive to Aurora A inhibitors [24,25,27,28]. Therefore, we wondered if an Aurora A-mediated activation of Polo kinase could also regulate mitotic progression in NSCs. We hypothesized that the expression of a constitutively active Polo kinase (Polo[T182D] in flies) should rescue the longer time in mitosis of *aurA* mutants NSCs. In order to test this idea, we generated new transgenic flies allowing the expression of different GFP-tagged variants of Polo kinase in brain tissues including wild type (Polo[WT]), kinase-dead (Polo[K54M]), constitutively active (Polo[T182D]) and constitutively inactive (Polo[T182A]). Live-imaging experiments on intact brains confirmed that each variant displayed a particular localization on centrosomes, mitotic spindle, midbody and kinetochores of NSCs, similar to what was previously reported in early embryos and S2 cells (S2 Fig and S1 Movie) and [29]. Moreover, only Polo[WT]::GFP was able to restore viability of the strong *polo*[9/10] hypomorph (hereafter referred to as *polo* mutant) when expressed under the control of a ubiquitous Actin5C promoter (Fig 2A and 2B). We then assayed the ability of each of these variants to reduce the time in mitosis of *polo* mutant NSCs by live-imaging microscopy (Fig 2C and 2D). Expression of Polo[T182D], Polo[T182A] and Polo[K54M] in a WT background slightly increased the time in mitosis of NSCs when compared to the expression of Polo[WT], suggesting hypermorphic or dominant-negative effects because these kinases variants harbor the Polo boxes that can displace endogenous Polo kinase [30]. Expression of Polo[WT] was able to restore a normal time in mitosis when expressed in *polo* mutants. Expression of the Polo[T182D] variant restored by more than half mitotic duration of *polo* mutant

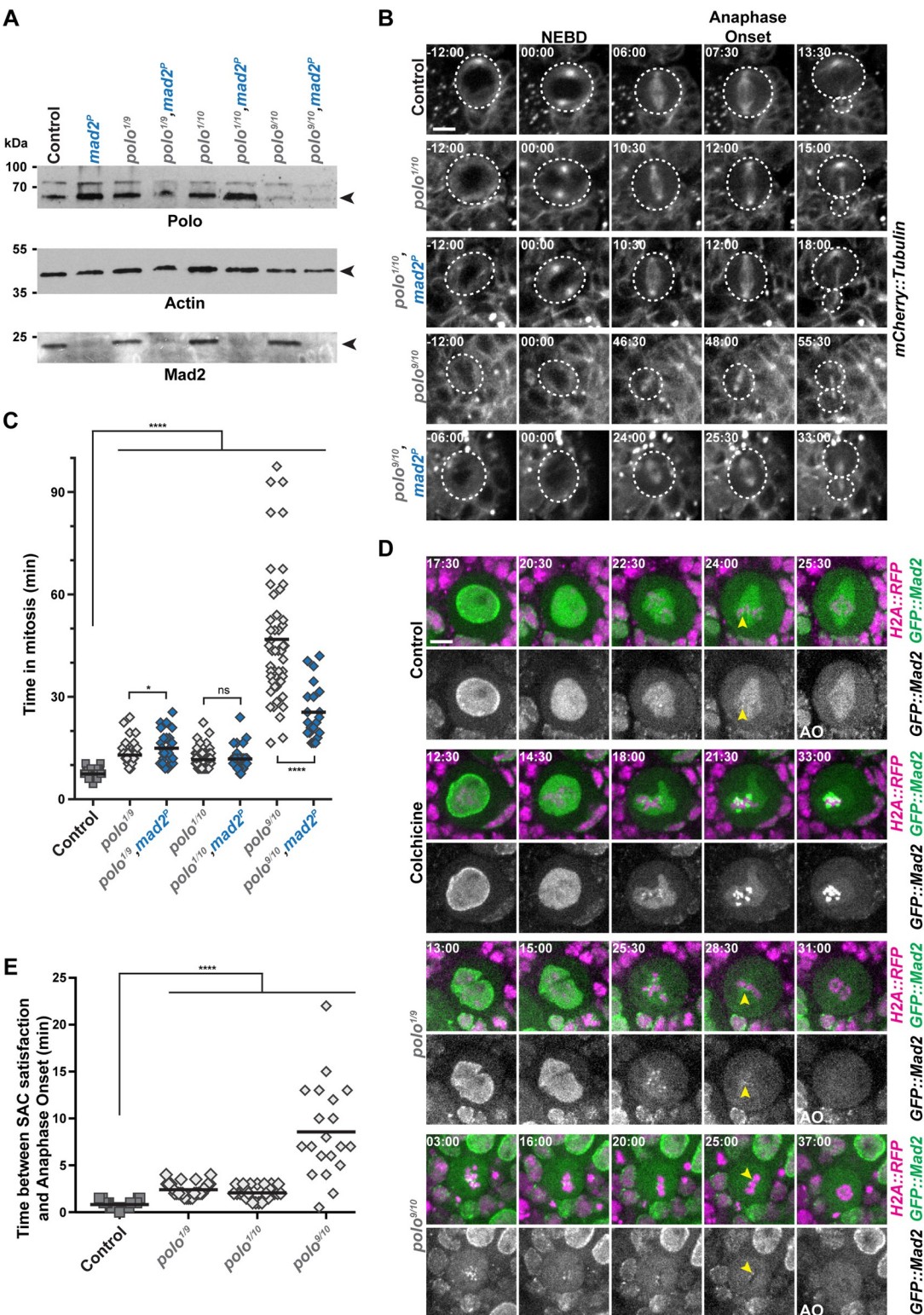

**Fig 1. *polo* mutant NSCs display a SAC-independent mitotic delay. A)** Polo, Mad2 and Actin Western blot analyses in *polo* and *polo,mad2$^P$* mutant brain extracts. The genotypes are indicated for each lane. *polo$^1$/polo$^{10}$* and *polo$^1$/polo$^9$* are weak hypomorphs and *polo$^9$/polo$^{10}$* is a strong hypomorph. **B)** Time-lapse imaging of mitosis in *polo* mutants in the absence or presence of Mad2. Selected image series of dividing NSCs of the indicated genotypes expressing mCherry::Tubulin. The white dashed line outlines the dividing NSCs. Scale bar: 5 μm. Time is min:s (t = 00:00 is NEBD). **C)** Quantification of time in mitosis

from NEBD to anaphase onset for NSCs of the indicated genotypes. Means are shown as black bars. Mann-Whitney unpaired test: ns: $p > 0.05$; *: $p < 0.05$; ****: $p < 0.0001$. **D)** GFP::Mad2 kinetochore localization in control, colchicine-treated, and *polo* mutant NSCs expressing H2A::RFP. Colchicine treatment (20μM) was used as a control for GFP::Mad2 accumulation on kinetochores. Yellow arrowheads indicate the last kinetochore labeled with GFP::Mad2 before anaphase onset (AO). Scale bar: 5 μm. Time is min:s (t = 00:00 is the beginning of the experiment). **E)** Quantification of the time between SAC satisfaction (disappearance of the last GFP::Mad2-labeled kinetochores) and anaphase onset in NSCs of the indicated genotypes. Means are shown as black bars. Mann-Whitney unpaired test: ****: $p < 0.0001$. Numerical data (mean, SD and *n*) related to all the figures are available in S1 Data.

NSCs. By contrast, expression of the inactive Polo^T182A or the Polo^K54M variants did not. It indicates that Polo kinase activity and more specifically activating phosphorylation on T182, is required for normal mitotic progression.

We envisaged that the expression of a constitutively active Polo^T182D variant may be detrimental for spindle assembly and trigger a SAC-dependent mitotic delay, as shown in previous studies [31–34]. We challenged this hypothesis by measuring the time in mitosis of NSCs upon Polo^T182D expression in *mad2^P* and *polo,mad2^P* double mutant (Fig 2E). The moderate increased time in mitosis resulting from Polo^T182D expression in WT cells was not abrogated following deletion of the SAC, indicating that this delay did not result from SAC activation but by constitutive Polo kinase activity. Congruently, expression of Polo^T182D was unable to fully restore the time in mitosis of *polo,mad2^P* double mutant NSCs. Altogether, this reveals that a fine-tuning of Polo activity thresholds is crucial to fulfill normal mitotic progression in a SAC-independent manner.

## Centrosomal AurA and centromeric Ial/AurB kinases are both required for Polo-dependent mitotic progression

We wanted to investigate if the mitotic progression defect observed in *aurA* and in *aurA, mad2^P* mutants shown in our previous study was Polo-kinase-dependent [2]. Thus, we expressed Polo^WT, Polo^T182A and Polo^T182D in these backgrounds (Figs 3A, 3B, S3A and S2 Movie). We found that expression of Polo^T182D was able to reduce the time in mitosis of both *aurA* and *aurA,mad2^P* mutants NSCs, suggesting that Polo kinase T182 phosphorylation by Aurora A contributes to mitotic progression. In addition, a study has shown that the mitotic phosphorylation of Polo on T182 is also mediated by Aurora B kinase, a component of the Chromosome Passenger Complex (CPC), at centromeres and during cytokinesis [23,29,35]. We therefore wondered if the fly orthologue of Aurora B, known as Ial (Ipl1-Aurora-like), was also contributing, with Aurora A, to the control of the time in mitosis through Polo kinase phosphorylation on T182. As described before for CPC mutants, NSCs from *ial* hypomorphic mutant displayed moderate prolonged mitosis (9.2 min for *ial^2A43/ial^1689* vs 6.0 min in a control, Figs 3C, 3D, and S3C and [36]). This delay was significantly reduced upon Polo^T182D but not by Polo^T182A expression indicating that the mitotic delay in *ial* mutant is partially dependent upon T182 phosphorylation. We found that combination of *ial* and *aurA* mutations triggered a synergistic delay in the time in mitosis, as most of the double mutant NSCs failed to exit mitosis after 150 min (S3B and S3C Fig). Expression of an active Polo^T182D did not recue the mitotic arrest observed in these double *aurA,ial* mutant background (S3C Fig).

Polo kinase activity is spatially regulated at the spindle poles and the centromeres by Aurora A and Aurora B respectively ([27,37] for review and [35]). To investigate loss of Aurora kinase activities with spatial activation of Polo, we first monitored the levels of T182 phosphorylation at the centromeres. We found that centromeric Ph-T182 Polo levels were high in control and *aurA* metaphase NSCs, and lowered in *polo* and *ial* mutants. We also found that *ial;aurA* double mutants NSCs that were arrested in mitosis displayed nearly normal levels of centromeric

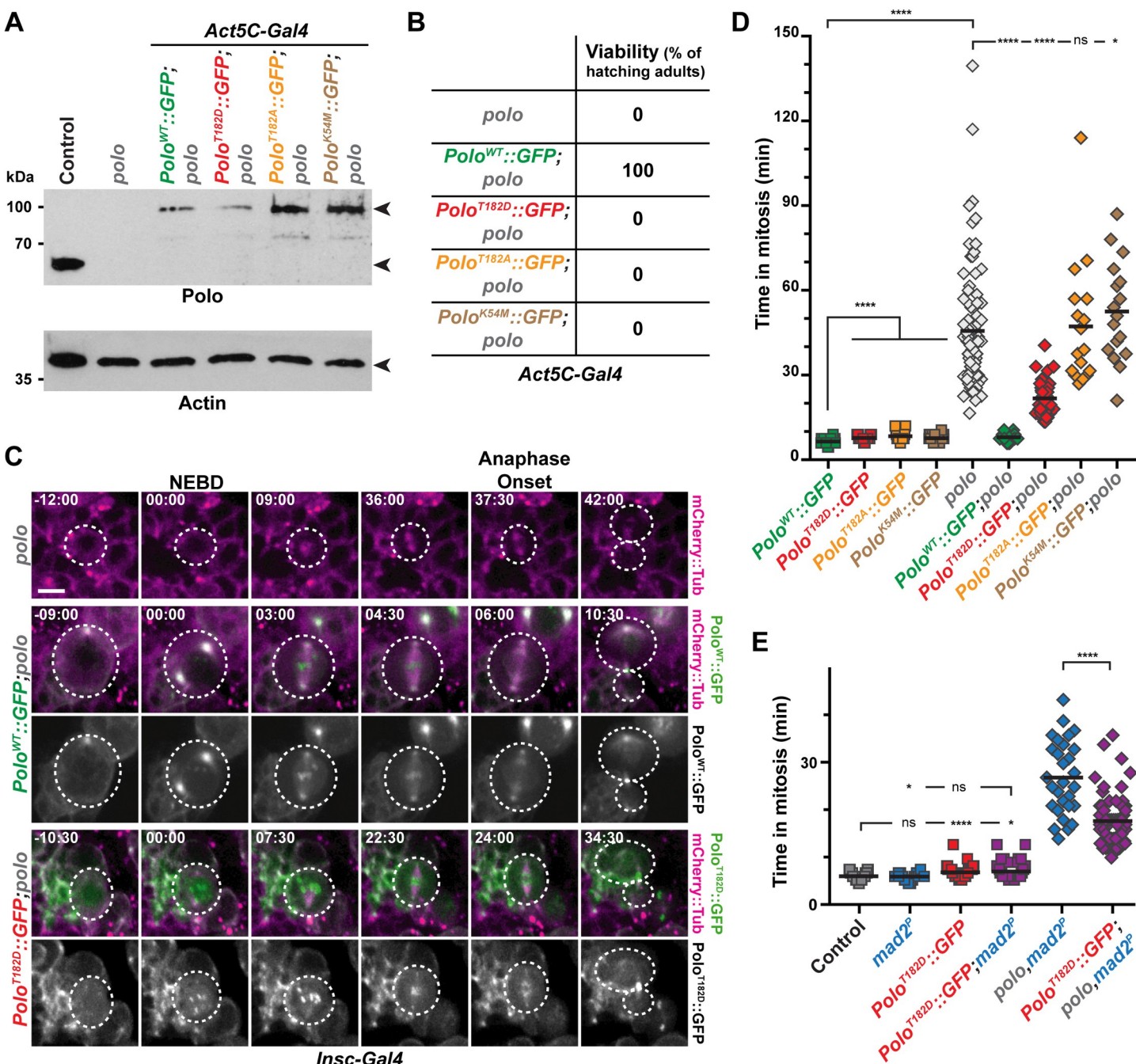

**Fig 2. Constitutive Polo kinase activity is not compatible with normal mitosis duration and *Drosophila* viability. A)** Polo and Actin Western blot analyses showing the expression of endogenous Polo kinase in control and GFP-tagged Polo variants (WT, T182D, T182A and K54M) expressed ubiquitously in *polo* mutant brains. The genotypes are indicated for each lane. Actin was used as a loading control. **B)** Hatching rates of *polo* mutants expressing the indicated Polo variants. **C)** Time-lapse imaging of mitosis in *polo* mutant NSCs expressing the indicated Polo variants and mCherry::Tubulin. The white dashed lines outline the dividing NSCs. Scale bar: 5 µm. Time is min:s (t = 00:00 is NEBD). **D-E)** Quantification of the time in mitosis of NSCs of the indicated genotypes. Means are shown as black bars. Mann-Whitney unpaired test: ns: $p > 0.05$; *: $p < 0.05$; ****: $p < 0.0001$.

Ph-T182 Polo, ruling out that centromeric active Polo is the main contributor that controls mitotic progression (S3D and S3E Fig). Because the anti- Ph-T182 Polo antibody did not label *Drosophila* mitotic centrosomes, we examined the levels of Centrosomin (Cnn) that

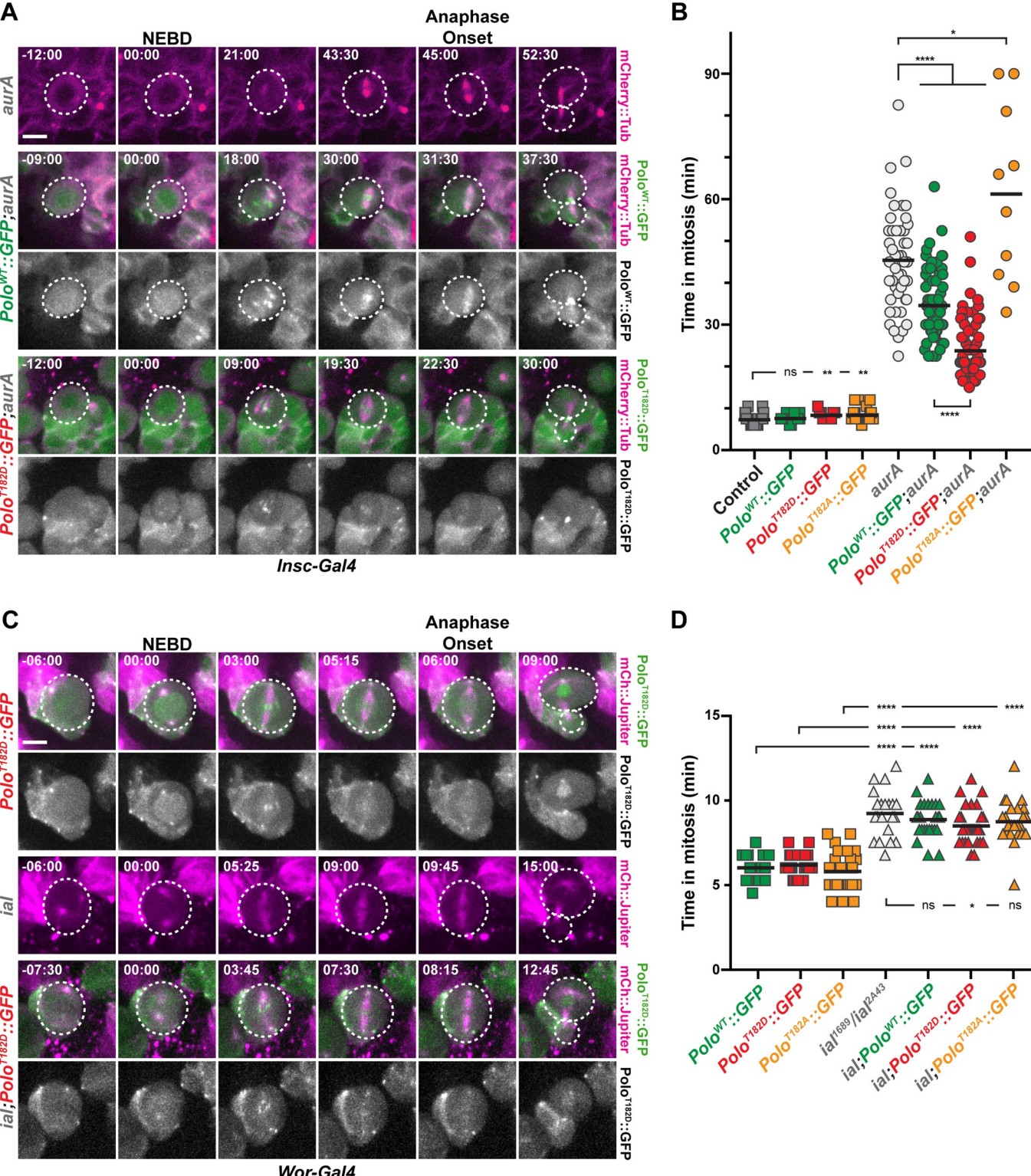

**Fig 3. Mitotic delays in *aurA* and *aurB* mutants are Polo-dependent. A, C)** Time-lapse imaging of mitosis in *aurA* (A) and *aurB/ial* (B) mutant NSCs expressing the indicated Polo variants and mCherry::Tubulin. The white dashed lines outline the dividing NSCs. Scale bar: 5 µm. Time is min:s (t = 00:00 is NEBD). **B, D)** Quantification of time in mitosis for the indicated genotypes. Means are shown as black bars. Mann-Whitney unpaired tests: ns: *p*>0.05; *: *p*<0.05; **: *p*<0.01; ****: *p*<0.0001.

oligomerizes and is recruited at centrosomes following direct phosphorylation by Polo kinase [38]. Compared to control metaphase NSCs, the levels of centrosomal Cnn were strongly reduced in *aurA* mutant. Centrosomal level of Cnn remained low in *aurA* mutant NBs expressing additional Polo$^{WT}$::GFP or the Polo$^{T182A}$ variant (S4 Fig). However, Polo$^{T182D}$ expression in mitotic aurA cells was able to significantly rescue the amount of centrosomal Cnn, although these levels were very low (only 10%) compared to those observed in control metaphase NBs. (S4A and S4B Fig). Altogether, these results strongly suggest that both Aurora A and Ial phosphorylates Polo on T182 to regulate mitotic progression of NSCs *in vivo*.

## Polo and the SAC protect NSCs against aneuploidy and excessive proliferation

Normal brain lobes are characterized by a restricted number of ~100 NSCs (reviewed in [7]). Defects in cell polarization or spindle orientation trigger the acquisition of a proliferating fate, the amplification of the NSC pool and ultimately tumor formation [39]. Because AurA and Polo are both required for cell polarization and mitotic spindle alignment in NSCs, *aurA* and *polo* mutant brains exhibit higher number of proliferating NSCs and induce tumors when injected in host flies [2,9,12,39,40]. To investigate if *polo* mutant-derived tumors share similar characteristics with *aurA* mutant, we undertook a careful analysis of the larval brain tumors derived from *polo* mutants. We found that NSC numbers increased in the central brain of weak *polo* mutants and ablation of the SAC with the *mad2$^P$* mutation enhanced this amplification (Fig 4A and 4B). As abnormal NSCs expansion is notoriously abrogated by aneuploidy in flies [2,3], we monitored the aneuploidy and polyploidy levels of weak *polo* hypomorphs associated with loss of the SAC (Fig 4C and 4D). We found that aneuploidy and polyploidy level, between 3 and 5% in weak *polo* hypomorphs, was largely increased upon ablation of the SAC (39 to 41%). It indicates that *polo* mutants are prone to segregation errors that are enhanced by loss of the SAC. Altogether, our data show that weak *polo,mad2$^P$* double mutants accumulate chromosome segregation and ploidy defects but exhibit higher NSC amplification. To check if these defects were correlated with changes in tissue proliferation, we performed brain transplantation experiments. We transplanted weak *polo* mutant brain tissues in which the SAC was active or not, in the abdomen of host WT flies and monitored the growth of the transplants as described before [39,41]. Strikingly, inactivation of the SAC and the subsequent increase in aneuploidy correlated with severely enhanced tumor growth of weaker *polo* mutant NSCs (45.1 and 51.5% vs. 22.0 and 24.7% of transplanted flies) (Fig 4E and 4F). SAC inactivation in a strong *polo* hypomorph background did not strongly impair NSC number (Fig 4A and 4B) nor tumor growth (85.2 vs. 86.5% of transplanted flies) (Fig 4E and 4F). These observations are similar to what was described with *aurA* mutant tumors [2]. Altogether our data suggest that Polo collaborates with the SAC to protect NSCs against chromosome segregation errors. Moreover, these results also reveal for the first time a case in which tumor growth is stimulated by aneuploidy in fly NSCs.

## AurA-dependent tumor suppression is mediated by Polo

Polo and AurA both are involved in mitotic progression and act like tumor suppressors in NSCs. Moreover, we noticed that cell size asymmetry resulting from the division of *aurA* mutant NSCs appeared to be rescued by Polo$^{T182D}$ expression (last time point in Fig 3A). Therefore, we decided to investigate if the tumor suppression potential of *aurA* mutants was also Polo-dependent. Consistent with several studies, the NSC number in *aurA* mutants was increased 10 to 25 times compared to control brains (1970 NSCs per lobe, Fig 5A and 5B) [2,9]. Expression of a Polo$^{WT}$ or the inactive Polo$^{T182A}$ variant did not inhibit NSC

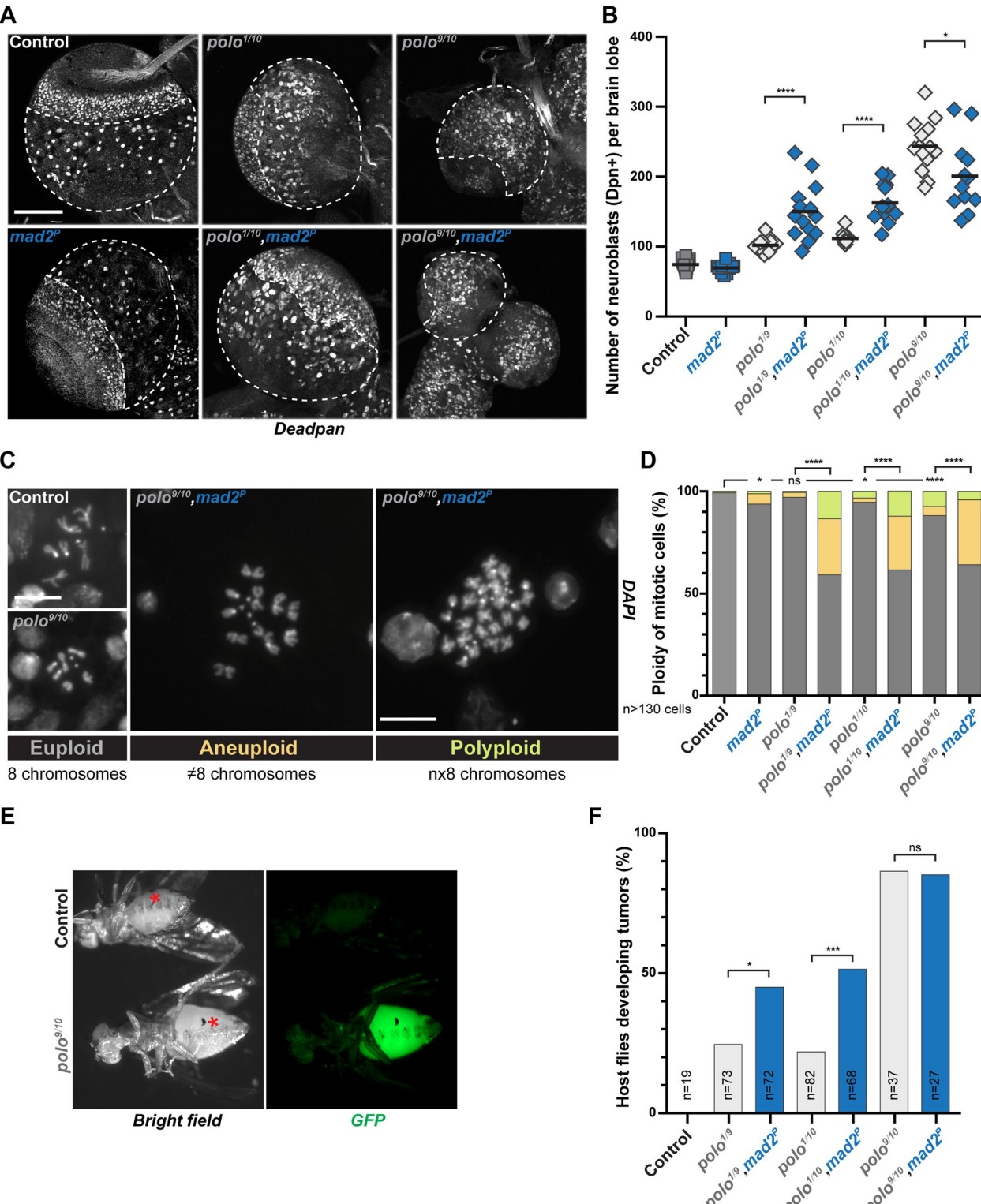

**Fig 4. Impairing the SAC in *polo* mutants induces aneuploidy and promotes tumor growth. A)** Representative images of larval brain lobes of the indicated genotypes stained for the NSC marker Deadpan. The white dashed lines outline the central brain area where NSCs were scored. Note the reduced brain lobe size in the *polo^{9/10}* mutant. Scale bar: 100 μm. **B)** Quantifications of Deadpan-positive NSCs per brain lobe for the indicated genotypes. Mann-Whitney unpaired test: *: *p*<0.05; ****: *p*<0.0001. Means are shown in black. **C)** Representative images of the three classes of mitotic figures observed from squashed brains stained with DAPI: euploid (8 chromosomes), aneuploid (more or less than 8 chromosomes) and

polyploid (multiple of 8 chromosomes) cells. Scale bar: 10 μm. **D)** Percentage of mitotic brain cells with ploidy defects (aneuploidy and polyploidy) for the indicated genotypes. For statistical analyses, the proportion of euploidy was compared to the proportion of ploidy defects. Fisher's exact test: ns: $p > 0.05$; *: $p < 0.05$; ****: $p < 0.0001$. **E)** Examples of wild-type adult host flies 15 days after transplantation with control and *polo* mutant brain lobe expressing H2A::GFP as a readout for tumor development (bottom). Asterisks indicate transplantation scars. **F)** Proportion of host flies developing tumors following the transplantation of brain lobes from the indicated genotypes. Fisher's exact test; ns: $p > 0.05$; *: $p < 0.05$; ***: $p < 0.001$.

amplification. However expression of the active Polo$^{T182D}$ was sufficient to strongly diminish the NSC amplification incurred in *aurA* mutant tissues (411 NSCs).

We also compared tumor growth progression after transplantation of *aurA* mutant neural tissues expressing or not Polo$^{T182D}$ (Fig 5C and 5D). We found that 90% of transplanted *aurA* mutant brain tissues labeled with mCherry produced large tumors that killed host flies shortly after 15 days. Expression of a Polo$^{WT}$ variant did not compromise the *aurA*-dependent strong tumor growth. However, when Polo$^{T182D}$ was expressed in *aurA* mutant, only 10% and 55% of the injected hosts developed tumors after 15 and 30 days respectively (Fig 5D). Importantly, the ploidy defects caused by *aurA* mutation (11.8% of the cells) were unchanged following Polo$^{T182D}$ expression (13.1% of the cells), ruling out differences in ploidy as the cause of the *aurA* mutant tumor growth restriction by Polo$^{T182D}$ (Fig 5E).

At the cellular level, Aurora A is required to regulate larval brain NSC number and growth by controlling mitotic spindle orientation and cell polarization, two characteristics of NSC asymmetric division [8,9,11,12,42]. We therefore analyzed these two processes in *aurA* mutant NSCs expressing Polo$^{T182D}$. We first monitored NSC polarization by analyzing the localization of aPKC and Miranda apical and basal crescents respectively. We found that both aPKC and Miranda crescents were properly localized in 100% of control or Polo$^{T182D}$ expressing NSCs. By contrast and in agreement with previous reports [9,42], most *aurA* mutant NSCs (78%) displayed homogenous cortical aPKC while Miranda was properly localized at the basal cortex (Fig 5F and 5H). Strikingly, expression of Polo$^{T182D}$ in *aurA* mutant restored the presence of the aPKC apical crescent in 84% of the NSCs but not expression of Polo$^{WT}$ or the Polo$^{T182A}$ variants. In parallel, we also measured mitotic spindle alignment along the apico-basal axis in *aurA* mutant with or without Polo$^{T182D}$ expression. In control and Polo$^{T182D}$ expressing NSCs, the vast majority of mitotic spindles were properly oriented in metaphase (angle < 30˚: 95% and 100%) while *aurA* mutant NSCs displayed a random mitotic spindle alignment (angle < 30˚: 41.3%). We found that Polo$^{T182D}$, expression was sufficient to significantly restore the *aurA* mutant-dependent spindle alignment defects (angle < 30˚: 76.3%; Fig 5F, 5I and 5J) while neither expression Polo$^{WT}$ or Polo$^{T182A}$ were able to rescue this defect. Altogether, these results indicate that Polo functions downstream of AurA to ensure the asymmetry of NSC divisions to prevent their excessive proliferation during larval brain development.

## Discussion

Tissue homeostasis in the fly larval brain requires timely mitotic progression for accurate chromosome segregation. Indeed, chromosome segregation defects trigger the formation aneuploid NSCs that fail to proliferate [2,3,17]. Mitotic state is regulated by the maintenance of Cyclin B-CDK1 activity and is under the control of the SAC and a functional APC/C. In a previous study, we showed that Aurora A kinase was also required for timed and efficient Cyclin B degradation in a SAC-independent manner [2]. We now bring in this new study details on the molecular mechanism involved: an Aurora-Polo signaling cascade is pivotal to control cell division length. Similar to *aurA*, *polo* mutants NCSs are characterized by a defective Cyclin B degradation. The loss of Aurora A or Ial/Aurora B leads to an extension of the time in mitosis. This mitotic delay can be rescued by the expression of a constitutively active Polo$^{T182D}$,

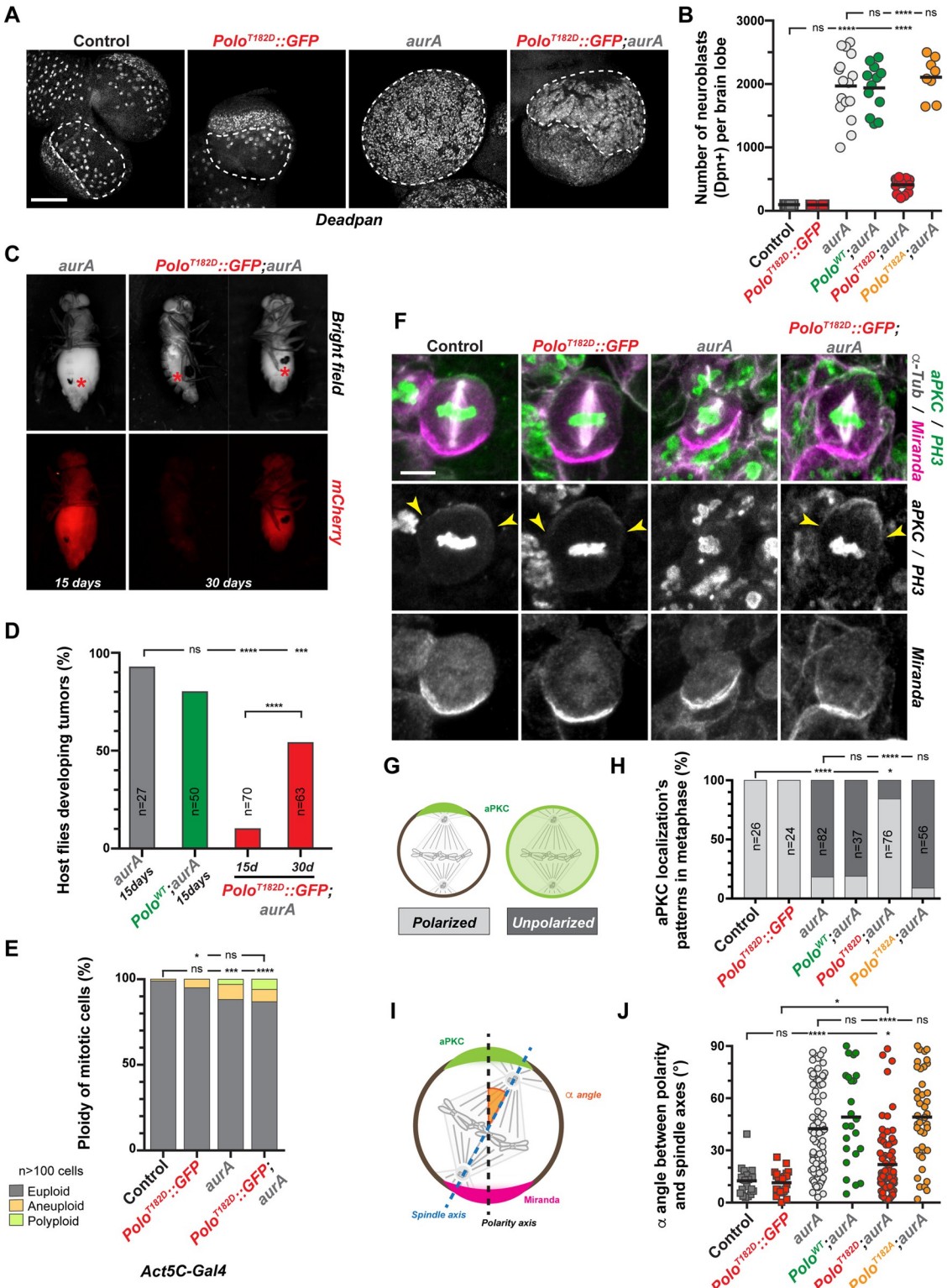

**Fig 5. Activation of Polo restores NSCs polarization, spindle orientation and inhibits tumor growth in an _aurA_ mutant. A)** Representative images of fixed control and _aurA_ mutant larval brain lobes with or without Polo[T182D]::GFP expression and stained for the NSC marker Deadpan. The white dashed lines outline the central brain area where NSCs were scored. Note that the expression Polo[T182D] in _aurA_ mutant restores the delimitation between the central brain and neuroepithelium. Scale bar: 100 μm. **B)** Quantification of the NSC number in each brain lobe for the indicated genotypes. **C)** Examples of wild-type adult host flies 15 and 30

days after transplantation with *aurA* mutant or *aurA* mutant expressing Polo$^{T182D}$::GFP brain lobes expressing mCherry::Tubulin as a readout to follow tumor development. Red asterisks indicate transplantation scars. **D)** Percentage of host flies developing tumors 15 and 30 days after the transplantation of brain lobes from the indicated genotypes. **E)** Quantification of ploidy defects scored in neural tissue for the indicated genotypes. The proportion of euploidy was compared to the proportion of ploidy defects for statistical analyses. **F)** Representative images of control, Polo$^{T182D}$::GFP, *aurA* mutant and Polo$^{T182D}$::GFP;*aurA* metaphase NSCs stained for Miranda (magenta in the merge), aPKC and Phospho-Histone H3 (green in the merge), and α-Tubulin (white in the merge). Yellow arrowheads indicate the edges of aPKC crescent. Scale bar: 5 μm. **G)** Schematic representation of aPKC localization patterns observed in NSCs: left: aPKC forms a crescent which defines the apical cortex of the control cell; right: aPKC is unpolarized and exhibits a homogenous cytoplasmic and cortical localization. **H)** Percentage of NSCs of the indicated genotypes with either a polarized apical crescent (light grey) or homogenous cortical and cytoplasmic aPKC localization (dark grey). **I)** Schematic illustration of the method used to measure the spindle alignment α angle (orange) between the polarity axis (black dashed line), determined by aPKC (green) and Miranda (magenta), and the spindle axis (blue dashed line) determined by spindle poles (See Material and Methods). **J)** Quantification of the α angle between polarity and spindle axes in metaphase NSCs for the indicated genotypes. Mann-Whitney unpaired test: ns: $p > 0.05$; ****: $p < 0.0001$. Means are shown as black bars. For panels D), E) and H): Fisher's exact test: ns: $p > 0.05$; *: $p < 0.05$; ***: $p < 0.001$; ****: $p < 0.0001$.

consistent with the idea that an Aurora-dependent phosphorylation and activation of Polo at T182 is the underlying connection. Our results suggest that although Ial contributes to mitotic progression, the pool of Polo phosphorylated by Ial does not appear to strongly regulate the time in mitosis because normal levels of centromeric Ph-T182 Polo are found in *aurA* mutant and in *ial;aurA* double mutant (S3D and S3E Fig). By contrast, mitotic progression is partly rescued in *aurA* mutant following expression of Polo$^{T182D}$ (Figs 3A, 3B and S3). In these conditions, we observed a weak restoration of Cnn recruitment on the centrosomes (as a readout of Polo activity at spindle poles). Altogether, our results favor a model in which centrosomal Aurora A phosphorylates Polo to produce enough active levels of kinase to support timely Cyclin B degradation and consequently cell division length.

Finally, although we have shown that Polo functions downstream of Aurora family kinases to regulate mitosis timing, an *aurA;ial* double mutant remains blocked in metaphase in the presence of the active Polo$^{T182D}$ mutant. It is therefore likely that other Aurora-dependent phosphorylation events are required to control Cyclin B degradation (See model in Fig 6). The use of weak or strong hypomorphic *polo* mutants and the controlled expression of a dominant Polo$^{T182D}$ active kinase allowed to establish the importance of Polo kinase activity thresholds for mitotic progression. Indeed, weak *polo* hypomorphs are characterized by near normal spindle formation and a prolonged mitosis (~13 min) in which the SAC has no contribution.

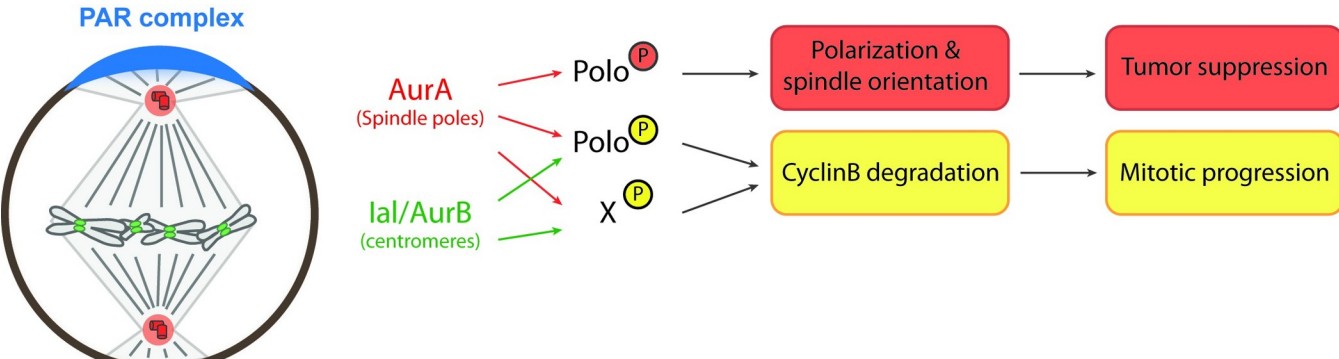

**Fig 6. Model showing Aurora(s)-dependent activation of Polo kinase for mitotic progression and tumor suppression in NSCs.** In fly NSCs, centrosomal/spindle pole-associated Aurora A kinase phosphorylates the Polo activating T182 of to generate active Polo kinase. This event is essential to regulate PAR complex polarized localization/activation and avoid tumor suppression. This activation is possible without centrosomes. To promote timely Cyclin B degradation and mitotic progression, Polo activation requires T182 phosphorylation by both spindle pole-Aurora A and centromeric Ial/Aurora B. This event occurs in parallel of the Spindle assembly checkpoint. Downstream Aurora A and Ial/Aurora B uncharacterized targets participate to this process.

Strong *polo* hypomorphs exhibit a longer mitotic delay (~47 min) that is also partly SAC-independent (Figs 1B, 1C, and S1). A past study showed that overexpression of an active Polo $^{T182D}$ triggers defective kinetochore attachments to spindle MTs [31]. The use of the same variant Polo$^{T182D}$ allowed highlighting another subversive function of Polo kinase in the control of cell division length. This reveals that Polo thresholds of activation are essential for timely Cyclin B degradation and mitotic exit *in vivo*. Our observations could therefore explain the apparent antagonism of the literature: Polo inhibits the APC/C activator Cdc20 when the SAC is unsatisfied but is also necessary to phosphorylate and activate APC/C to trigger mitotic exit [43–45].

Tissue homeostasis depends on the asymmetric distribution of cell fate determinants during NSC division that prevents their excessive proliferation. This process is regulated by tumor suppressor genes [4,5,7,46]. Polo and Aurora A protein kinases are both required for cell polarization in the one-cell *C. elegans* embryos and in *Drosophila* NSCs [9,12,42,47–50]. However, this work demonstrates a direct molecular and functional link between these two kinases in the context of organogenesis and tumor suppression. Our study shows that Polo phosphorylation on T182 by Aurora A governs polarization of neural stem cells but also mitotic spindle orientation, two processes that are essential for appropriate cell fate acquisition during asymmetric cell division. Whether this activation is locally and solely regulated at the centrosome remain unclear. Indeed, centrosomes appear to remain poorly functional in *aurA* mutant NBs expressing the active Polo$^{T182D}$ variant. Furthermore, *sas-4* mutants NSCs lacking centrosomes appear to display normal cortical polarity [51]. We therefore favor the hypothesis that providing enough overall active Polo kinase levels may be sufficient to polarize NSCs. The local centrosome activation at spindle poles may not be an absolute requirement for this process. In agreement with this hypothesis, mutation in the Aurora A activator Bora, triggers polarity defects but Bora is cytoplasmic during cell division, and does not show centrosomal or spindle pole localization [20,52,53].

A past study has proposed that direct phosphorylation of the PAR complex by Aurora A contributes to cell polarization [11]. We show here that the amplification of NSCs and the tumor growth of *aurA* mutant brains are strongly inhibited by the expression of an active Polo variant. This restoration of tissue homeostasis is caused by the near complete rescue of cell polarity defects. We therefore suggest that Polo kinase phosphorylation of yet unidentified substrate(s) mediate polarization of fly NSCs. PAR3 is regulated by direct PLK-1 phosphorylation in the *C.elegans* embryo although the effects on embryonic polarization are unclear [49]. Further studies will be required to investigate if this direct regulation event is also conserved in fly NSCs. Our results also suggest that Aurora A phosphorylation of the PAR complex may only play a minor role for Aurora A-dependent tumor suppression.

Rescue of spindle orientation in *aurA* mutant by expression of the active Polo$^{T182D}$ may also be mediated by a direct regulation of the spindle orientation machinery by Polo. However, spindle orientation may also be indirect and stimulated by the weak centrosomal maturation (S4 Fig), triggering microtubule nucleation and interactions with the cell cortex [22,37,38]. In conclusion, this work has revealed that two essential characteristics of NSCs, the progression of mitosis mediated by efficient Cyclin B degradation and cell polarization, are coupled by a signaling cascade leading to the activation of the Polo protein kinase by Aurora kinases. This Aurora/Polo module is critical for tissue homeostasis and when impaired, triggers NSC amplification and tumor development. It is also important to emphasize that our study has shown that neural tumors from *polo* mutants seem to have specific characteristics: their progression is aggravated by aneuploidy. Further studies will be required to identify and investigate the functions of the Polo targets in these essential processes and how they are regulated in space and time.

## Materials and methods

### Fly strains and alleles combinations

The following strains were used for this study:

UAS-mCherry::Tubulin (BDSC #25774), Ubi-RFP::Tubulin (gift from Renata Basto), UASt-mCherry::Jupiter (gift from Clemens Cabernard), UAS, His2Av::mRFP1 (H2A::RFP, BDSC #23651), His2Av::GFP (H2A::GFP) [54] and GFP::Mad2 [26] were used to monitor by live-imaging the microtubule network, chromosomes and unattached kinetochores. Wild-type (UASp-Polo$^{WT}$::GFP), phosphomutant (UASp-Polo$^{T182A}$::GFP), phosphomimic (UASp-Polo$^{T182D}$::GFP) and kinase-dead (UASp-Polo$^{K54M}$::GFP) Polo variants tagged with GFP were expressed using the Gal4-UAS system. All four constructs were inserted on the second chromosome (attP40, see below) and on the third chromosome (attP154, [29]). Cyclin B::GFP expressing flies were described before [55].

Inscuteable-Gal4 (Insc-Gal4; BDSC #8751) and Worniu-Gal4 (Wor-Gal4; BDSC #56554) allowed the expression of UAS driven fluorescent markers in the neuroblasts; and Actin5C-Gal4 (BDSC #4414) was used for ubiquitous expression in somatic cells. The expression of Polo variants for protein level and rescue assays in Fig 2A and 2B as well as the expression of Polo$^{T182D}$::GFP for ploidy analysis (Fig 5E) were performed using the Actin5C-Gal4 driver. The expression of mCherry::Jupiter and Polo$^{T182D}$::GFP in *ial* mutant background were driven by Wor-Gal4. For other experiments (live-imaging, immunofluorescence and transplantation assay), Polo variants and the microtubule marker were expressed under the control of Insc-Gal4.

The following alleles were used in this study and described before: *aurA$^{8839}$* (referred as *aurA* mutant, [9,40]), *mad2$^p$* [26], *polo$^1$* [56], *polo$^9$* and *polo$^{10}$* [57], *ial$^{1689}$*, *ial$^{35.33}$* and *ial$^{2A43}$* [36]. The allelic combination *polo$^9$/polo$^{10}$* (referred as *polo* mutant) corresponds to a strong *polo* hypomorph mutant and the flies do not reach the pupal stage [57]. The allelic combinations *polo$^1$/polo$^9$* and *polo$^1$/polo$^{10}$* mutant flies are viable and correspond to weak *polo* hypomorphic mutants. The phenotype of Ial/Aurora B loss-of-function was assessed during the larval stage using two heteroallelic combinations: *ial$^{35.33}$/ial$^{1689}$* and *ial$^{2A43}$/ial$^{1689}$*.

Flies were maintained and crossed under standard conditions at 25˚C.

### Generation of transgenes

The constructs containing the coding sequence of Polo variants (WT, T182A, T182D and K54M) C-terminally tagged with GFP were cloned into the plasmid pUAS-K10-attB [29] and were injected in embryos for targeted insertion on the attP40 site located on second chromosome (BestGene Inc.).

### Immunohistochemistry and antibodies

Third instar larval brains (120 to 144 h after egg lying depending on overall growth delay) were dissected in Schneider medium supplemented with 10% FCS and fixed in Testis Buffer (183 mM KCl, 47 mM NaCl, 10 mM Tris, and 1 mM EDTA, pH 6.8) containing 10% formaldehyde and 0.01% Triton X-100 at 25˚C for 20 minutes, as described in a previous study [7].

Primary antibodies: monoclonal anti-α-tubulin DM1A (mouse, 1:500, Sigma T6199), anti-PKCζ (C-20) (rabbit, 1:250, Santa Cruz sc-216), anti-Miranda (rat, 1:200, Abcam 197788), polyclonal anti-phosphorylated Ser10 of Histone H3 (rabbit, 1:500, Millipore 06–570), anti-Deadpan (rat, 1:200, Abcam 195172), anti-Cnn (rabbit, 1:500, [58]) and monoclonal anti-phospho-Plk1T210 (mouse, 1:500, Abcam ab39068).

Secondary antibodies: anti-rat, anti-mouse and anti-rabbit conjugated to 488, 568, 647 Alexa-Fluor (1:500 to 1:1000, Life Technologies).

Images were acquired on a Leica SP5 confocal microscope using 40X (1.25 N.A.) and 63X (1.40 N.A.) objectives, controlled with Leica LAS software and analyzed on Fiji or Imaris.

## Western blotting and antibodies

Two to three brains were dissected, resuspended in Laemmli Buffer and loaded on a 4–15% Mini-PROTEAN TGX Precast Protein Gel (BioRad). After migration, proteins are transferred on a nitrocellulose membrane using a Trans-Blot Turbo (BioRad). Membranes were washed with PBS+0.1% Tween20 (PBST) and blocked with PBST+5 to 10% powder milk. The membranes were then incubated in the following primary antibodies, diluted in TBST+2.5 to 5% powder milk: Rabbit polyclonal anti-Actin (I-19, Santa Cruz sc-1616-R; 1/4000), anti-Mad2 (Gift from David Sharp; 1/500), mouse monoclonal anti-Polo MA294 (Gift from David Glover; 1/100). Anti-mouse and anti-rabbit secondary antibodies were purchased from Jackson ImmunoResearch. Western blots were revealed with SuperSignal West Dura or Femto (ThermoFisher).

## Live cell imaging

Third instar larval brains (120 to 144h after egg lying depending on overall growth delay) were dissected in Schneider medium containing 10% FCS and transferred to 50 μL wells (Ibidi, μ-Slide Angiogenesis) for live imaging. Mutant and control brains were imaged in parallel at 25˚C. Z-series (thickness of 20 μm with 1 μm spacing) were acquired with a temporal resolution of 30 to 90 s for 1 to 2.5 hours. Alternatively, samples were mounted on a stainless-steel slide, between coverslip and mineral oil as described in a previous study [59].

Images were acquired with a spinning disk system consisting of a DMi8 microscope (Leica) equipped with a 63X (1.4 N.A.) oil objective, a CSU-X1 spinning disk unit (Yokogawa) and an Evolve EMCCD camera (Photometrics). The microscope was controlled by the Inscoper Imaging Suite and the dedicated software (Inscoper). Alternatively, a CSU-X1 spinning-disk unit mounted on an inverted microscope (Elipse Ti; Nikon) equipped with a 60X (1.4 N.A.) oil objective, a sCMOS ORCA Flash 4.0 (Hamamatsu) and controlled by MetaMorph; was also used for some experiments. Images were processed with Fiji or Imaris softwares.

## Image analyses

**Time in mitosis.**   The time in mitosis was determined as described in [31,59,60]. Briefly, the mitotic entry point was determined when an increase of the mCherry::tubulin signal was detected in the nuclear region, reflecting dismantlement and permeabilization of the nuclear envelope to tubulin. Anaphase onset was determined as the first spindle elongation time point.

Note that only the cells undergoing complete mitosis within the span of the experiments were scored. For *aurA;ial* double mutant (S3B and S3C Fig) displaying the most severe phenotype, the cells could not complete mitosis throughout the experiment (2.5 h) and the time in mitosis could not be measured.

**Cyclin B::GFP degradation.**   Cyclin B::GFP degradation kinetics were analyzed in individual NSCs undergoing complete mitosis within the span of the experiments. GFP total fluorescence intensities were quantified for the whole cell. The signal was adjusted for background and for bleaching relative to the signal of a non-mitotic cell. In the graphs, signal levels for the whole cell are displayed as normalized signals relative to the maximal intensity measured for the cell as described in previous studies [2,26,61].

**Spindle orientation.** Spindle orientation on fixed tissue was measured on metaphase cells using the angle tool of Fiji or alternatively using the "spot" tool of Imaris. Briefly, two spots were placed in the center of the apical aPKC and basal Miranda crescents to determine the polarity axis; two spots positioned at the spindle poles determined the mitotic spindle axis. For *aurA* mutant that did not exhibit an apical aPKC crescent, the polarity axis was determined by a straight line between the center of the Miranda crescent and a second spot at the opposite cortex. The 3D coordinates of the spots were used to calculate polarity and spindle axes vectors and the angle between them as described in [62].

**Cnn intensity.** Total Cnn total intensity ("Sum slices") was measured on spindle poles of metaphase NSCs using Fiji software. The spindle poles were defined thanks to Cnn and tubulin staining. For each cell, 1 measurement in the cytoplasm was used and subtracted from the spindle pole intensity values.

**Phospho-Plk1 intensity.** Z-projections of images restricted to the metaphase plates were made using Fiji ("Sum slices"). Using phospho-Histone H3 staining, metaphase plate shapes were drawn on Fiji (orange lines in S3D Fig) and phospho-Plk1 total signal intensities were measured. Raw intensities were normalized by the volume of measurement.

## Ploidy analysis in brains

Third instar larval brains were dissected in PBS, incubated for 8 minutes in 0.5% sodium citrate solution, fixed in acetic acid (45% for 45 seconds and 60% for 2 minutes), squashed between slide and coverslip and flash-frozen in liquid nitrogen [2]. After coverslip flip off, the squashed cells on the slide were washed in PBS and mounted in ProlonGold supplemented with DAPI. Images were acquired using a Leica DMRXA2 microscope equipped a 63X (1.32 N.A.) objective, a CoolSnap HQ2 camera and controlled by MetaMorph. Images were processed with Fiji software.

## Tumor transplantation assay

The protocol to assess proliferation ability of brain tissues was originally set up in the lab of Cayetano Gonzalez [41]. Briefly, a central brain fragment (expressing a fluorescent probe) was transplanted in the abdomen of a live wild-type adult female fly and the growth of the transplant was monitored over time. Control or mutant third instar larval brains, expressing either His2Av::GFP or UAS-Cherry::Tubulin under the control of Insc-Gal4 were dissected in sterile PBS (Sigma). One brain lobe was transplanted in the abdomen of a wild-type adult fly using a pulled capillary with a beveled tip (around 150 μm diameter) adjusted to a microinjection system (IM-9B; Narishige). After transplantation, host flies were maintained at 18°C overnight and transferred at 25°C for a month. Tumor growth was monitored every one or two days upon 30 days for the appearance of a mCherry or GFP signal in the abdomen. For *aurA* and strong *polo* mutants, transplant growth was assessed over a 15-days period because these tumors were particularly aggressive and killed the host flies before 30 days. The neuroblast-specific Insc-Gal4 was used to assay the effect of Polo$^{T182D}$::GFP variant for tumor growth.

## Statistical analyses

Sample size and statistical test used for each analysis are found in the corresponding figure legends. Statistical analyses were performed using the Prism software (Version 7, GraphPad). Three tests were used according to the data. To compare ranks between two unpaired samples with variable variances and non-Gaussian distributions, a nonparametric Mann-Whitney test was used. To compare two unpaired samples with Gaussian distribution, a nonparametric t test was used complemented with a Welch correction if standard deviations were unequal (F

test). The comparisons of proportions were assessed by a two-sided Fisher's exact test with a confidence index of 95%. Significance are displayed on the figure: ns: $p > 0.05$; *: $p < 0.05$; **: $p < 0.01$; ***: $p < 0.001$; ****: $p < 0.0001$. Numerical data (mean, SD and $n$) related to all the figures are available in S1 Data.

## Supporting information

**S1 Fig. Cyclin B degradation kinetics in *polo* mutant NSCs. A)** Time-lapse imaging of mitosis in *polo* mutants in the absence or presence of Mad2. Selected image series of dividing NSCs of the indicated genotypes expressing mCherry::Tubulin (purple in the top panels and in the middle monochrome panels), and CyclinB::GFP (green and lower monochrome panels). The white dashed line outlines the dividing NSCs. Scale bar: 5 μm. Time is min:s (t = 00:00 is NEBD). **B)** Cyclin B::GFP degradation profiles in WT, in *polo⁹/polo¹⁰* NSCs, and in *polo⁹/polo¹⁰, mad2ᴾ* double-mutant NSCs. **C)** Quantification of the time in mitosis (min) in NSCs for the indicated genotypes. **D)** Quantification of the time (min) required for 50% of Cyclin B::GFP degradation for the indicated genotypes. Mann-Whitney unpaired tests: ns: $p > 0.05$; *: $p < 0.05$; ****: $p < 0.0001$.
(TIF)

**S2 Fig. Localization of GFP-tagged Polo variants during NSC division. A)** Time-lapse imaging of cell division in NSCs expressing the indicated Polo variants and mCherry::Tubulin. The white dashed line outlines the NSCs. Scale bar: 5 μm. Time is min:s (t = 00:00 is NEBD). **B)** Summary table of Polo variants localization patterns along cell cycle stages. The symbols "-", "+/-", "+" and "++" reflect absence, variable localization, moderate localization and strong enrichment, respectively.
(TIF)

**S3 Fig. Analysis Polo activation in different mutant backgrounds. A)** Quantification of the time in mitosis for the indicated genotypes. **B)** Time-lapse imaging of dividing NSCs for control and *ial;aurA* double mutant NSCs expressing RFP::Tubulin. The white dashed lines outline the dividing NSCs. Scale bar: 5 μm. Time is h:min:s (t = 0:00:00 is the beginning of the experiment). **C)** Quantification of the time in mitosis of NSCs of the indicated genotypes. The *ial;aurA* double mutant NSCs displayed a severe mitotic delay that could not be quantified since the vast majority did not complete mitosis during the duration of the experiment (2.5 hours). Mann-Whitney unpaired test: ns: $p > 0.05$; ****: $p < 0.0001$. **D)** Representative images of metaphase NSCs of the indicated genotypes stained for phospho-Plk1 (white in the merge), aPKC and Phospho-Histone H3 (magenta in the merge) and tubulin (green in the merge). The orange shapes in the right column outline the metaphase plates, according to the Phospho-Histone H3-labeled metaphase plate, in which phospho-Plk1 signal was measured. Scale bar: 5 μm. **E)** Quantification of phospho-Plk1 signal for the indicated genotypes, normalized by the area of measurement. Unpaired t-test: ns: $p > 0.05$; **: $p < 0.01$; ***: $p < 0.001$; ****: $p < 0.0001$.
(TIF)

**S4 Fig. Polo activation restores centrosome maturation in *aurA* mutant NSCs. A)** Representative images of metaphase NSCs of the indicated genotypes stained for Miranda (magenta in the merge), aPKC and Cnn (green in the merge) and tubulin (white in the merge). The yellow arrowheads in the bottom line highlight the Cnn-labeled spindle poles. Scale bar: 5 μm. B) Quantification of normalized Cnn intensity at spindle poles for the indicated genotypes. Mann-Whitney unpaired tests: ns: non-significant, *: $p < 0.05$; **: $p < 0.01$; ****: $p < 0.0001$.
(TIF)

**S1 Data. Numerical data corresponding to Figs 1, 2, 3, 4, 5, S1, S3 and S4.**
(XLSX)

**S1 Movie. Polo variants expressed in polo mutant NSCs larval NSCs of the indicated genotypes** (from left to right: wild-type, *polo* mutant, *polo* mutant expressing PoloWT:: GFP and polo mutant expressing Polo$^{T182D}$::GFP) expressing mCherry::Tubulin under the control of Insc-Gal4. Microtubule marker and Polo variants are shown in magenta and green respectively. Polo$^{WT}$ is able to restore a normal mitotic timing in *polo* mutant while Polo$^{T182D}$ decreases the mitotic delay by more than half. Scale bar: 5 μm. Time is min:s. « 00:00 » corresponds to NEBD.
(AVI)

**S2 Movie. Constitutively active Polo reduces the mitotic delay of aurA mutant NSCs.** Larval NSCs of the indicated genotypes (from left to right: wild-type, *aurA* mutant and *aurA* mutant expressing Polo$^{T182D}$::GFP) expressing mCherry::Tubulin under the control of Insc-Gal4. Microtubule marker and Polo variants are shown in magenta and green respectively. Polo$^{T182D}$ is able to reduce the time in mitosis of *aurA* mutant NSCs. Scale bar: 5 μm. Time is min:s. « 00:00 » corresponds to NEBD.
(AVI)

## Acknowledgments

We thank members of the "Cytoskeleton and Cell Proliferation" lab, Roland Le Borgne and Damien Coudreuse for helpful discussions. We are grateful to Renata Basto for advices on tumor growth assays, Anne Royou, Jordan Raff, Clemens Cabernard, David Sharp, Timothy Megraw, Juliette Mathieu and Jean-René Huynh for antibodies, flies and plasmids. We also acknowledge Stéphanie Dutertre and Xavier Pinson for their help using the microscopes of the Microscopy Rennes imaging center facility (MRic).

## Author Contributions

**Conceptualization:** Emmanuel Gallaud, Laetitia Bataillé, Régis Giet.

**Formal analysis:** Emmanuel Gallaud, Laurent Richard-Parpaillon, Laetitia Bataillé, Mathieu Métivier, Régis Giet.

**Funding acquisition:** Régis Giet.

**Investigation:** Emmanuel Gallaud, Laurent Richard-Parpaillon, Laetitia Bataillé, Aude Pascal, Mathieu Métivier, Régis Giet.

**Methodology:** Emmanuel Gallaud, Laurent Richard-Parpaillon, Laetitia Bataillé, Aude Pascal, Mathieu Métivier, Régis Giet.

**Project administration:** Régis Giet.

**Resources:** Régis Giet.

**Supervision:** Régis Giet.

**Validation:** Emmanuel Gallaud, Laurent Richard-Parpaillon, Laetitia Bataillé, Aude Pascal, Mathieu Métivier.

**Visualization:** Emmanuel Gallaud, Laurent Richard-Parpaillon, Laetitia Bataillé, Mathieu Métivier, Régis Giet.

**Writing – original draft:** Emmanuel Gallaud, Vincent Archambault, Régis Giet.

**Writing – review & editing:** Emmanuel Gallaud, Laetitia Bataillé, Vincent Archambault, Régis Giet.

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
