## [Editor Report · Decision Letter 0]

13 Sep 2021

Dear Dr Giet,

Thank you very much for submitting your Research Article entitled The spindle assembly checkpoint and the spatial activation of Polo kinase determine the duration of cell division and prevent neural stem cells tumor formation' to PLOS Genetics.

The manuscript was fully evaluated at the editorial level and by independent peer reviewers at Review Commons. The reviewers all considered your paper to be potentially interesting, but Reviewer #1 and #2 raised some substantial concerns about the current manuscript. Main questions need to be resolved include whether the WT and the putative phospho-mutant forms (T182A) of Polo rescue AurA mutant phenotypes and whether measurement of mitotic duration using alternative approach(s) other than mCherry-Tubulin yield similar results. Here, a Cyclin B-GFP reporter or fly FUCCI can be used as alternative approaches to measure mitotic duration. Based on the reviews and your point-by-point response to the reviewers’ comments, we will not be able to accept the current version of the manuscript, but we would be happy to consider a revised version.

If you decide to revise the manuscript for further consideration at PLOS Genetics, please aim to resubmit within the next 60 days, unless it will take extra time to address the concerns of the reviewers, in which case we would appreciate an expected resubmission date by email to plosgenetics@plos.org.

[LINK]

We are sorry that we cannot be more positive about your manuscript at this stage. Please do not hesitate to contact us if you have any concerns or questions.

Yours sincerely,

Yan Song, Ph.D.

Associate Editor

PLOS Genetics

Gregory P. Copenhaver

Editor-in-Chief

PLOS Genetics

---

## [Decision Letter · Decision Letter 1]

28 Feb 2022

Dear Dr Giet,

Thank you very much for submitting your revised manuscript to PLOS Genetics. I am pleased to let you know that, based on your revisions and the reviewer comments below, your manuscript can now be accepted in principle at PLOS Genetics. As you will see there are still some minor points raised by Reviewer #2 with regards to citation and typos. We will be able to formally accept your manuscript once you have uploaded final files that fully address these specific points made by Reviewer #2.

1) Provide a list of your responses to the review comments and a description of the changes you have made in the manuscript.

We hope to receive your revised manuscript within the next 1-2 weeks. If you anticipate any delay in its return, we would ask you to let us know the expected resubmission date by email to plosgenetics@plos.org.

[LINK]

Please let us know if you have any questions while making these minor revisions. We look forward to hearing from you.

Yours sincerely,

Yan Song, Ph.D.

Associate Editor

PLOS Genetics

Gregory P. Copenhaver

Editor-in-Chief

PLOS Genetics

Reviewer's Responses to Questions

**Comments to the Authors:**

Reviewer #1: The authors have adequately addressed comments from previous reviewers. The manuscript is now recommended for publication in PLoS Genetics.

Reviewer #2: The authors have added quantification of WT and T182A alleles in Figures 5 and S4 and have substantiated their conclusion that Cyclin B degradation is slowed in polo mutants. These additions have strengthened the manuscript and I am supportive of publication. Below I list 2 minor typos that should be corrected.

1) The section describing the Figure 2E should include an explanation of the T182A and K54M phenotypes, and should reference the papers showing attributing this to dominant negative effects.

2) Figure S1C. Y axis label should read “mitosis”

---

## [Editor Report · Decision Letter 2]

14 Mar 2022

Dear Dr Giet,

We are pleased to inform you that your manuscript entitled "The spindle assembly checkpoint and the spatial activation of Polo kinase determine the duration of cell division and prevent tumor formation" has been editorially accepted for publication in PLOS Genetics. Congratulations!

Yours sincerely,

Yan Song, Ph.D.

Associate Editor

PLOS Genetics

Gregory P. Copenhaver

Editor-in-Chief

PLOS Genetics

**Data Deposition**

http://datadryad.org/submit?journalID=pgenetics&manu=PGENETICS-D-21-01184R2

**Press Queries**

---

## [Editor Report · Acceptance letter]

30 Mar 2022

PGENETICS-D-21-01184R2 

The spindle assembly checkpoint and the spatial activation of Polo kinase determine the duration of cell division and prevent tumor formation 

Dear Dr Giet, 

We are pleased to inform you that your manuscript entitled "The spindle assembly checkpoint and the spatial activation of Polo kinase determine the duration of cell division and prevent tumor formation" has been formally accepted for publication in PLOS Genetics! Your manuscript is now with our production department and you will be notified of the publication date in due course.

With kind regards,

Zsofia Freund

PLOS Genetics

On behalf of:
